# Superior Wear Resistance in EBM-Processed TC4 Alloy Compared with SLM and Forged Samples

**DOI:** 10.3390/ma12050782

**Published:** 2019-03-07

**Authors:** Weiwen Zhang, Peiting Qin, Zhi Wang, Chao Yang, Lauri Kollo, Dariusz Grzesiak, Konda Gokuldoss Prashanth

**Affiliations:** 1Guangdong Key Laboratory for Processing and Forming of Advanced Metallic Materials, South China University of Technology, Guangzhou 510640, China; mewzhang@scut.edu.cn (W.Z.); 201720101729@mail.scut.edu.cn (P.Q.); cyang@scut.edu.cn (C.Y.); 2National Engineering Research Center of Near-net-shape Forming for Metallic Materials, South China University of Technology, Guangzhou 510640, China; 3Department of Mechanical and Industrial Engineering, Tallinn University of Technology, Ehitajete tee 5, 19086 Tallinn, Estonia; lauri.kollo@taltech.ee (L.K.); prashanth.konda@taltech.ee (K.G.P.); 4Department of Mechanical Engineering and Mechatronics, West Pomeranian University of Technology, Aleja Piastów 17, 70-310 Szczecin, Poland; dariusz.grzesiak@zut.edu.pl; 5Erich Schmid Institute of Materials Science, Austrian Academy of Sciences, Jahnstraße 12, A-8700 Leoben, Austria

**Keywords:** Ti-6Al-4V, wear, additive manufacturing, properties

## Abstract

The wear properties of Ti-6Al-4V alloy have drawn great attention in both aerospace and biomedical fields. The present study examines the wear properties of Ti-6Al-4V alloy as prepared by selective laser melting (SLM), electron beam melting (EBM) and conventional forging processes. The SLM and EBM samples show better wear resistance than the forged sample, which correlates to their higher hardness values and weak delamination tendencies. The EBM sample shows a lower wear rate than the SLM sample because of the formation of multiple horizontal cracks in the SLM sample, which results in heavier delamination. The results suggest that additive manufacturing processes offer significantly wear-resistant Ti-6Al-4V specimens in comparison to their counterparts produced by forging.

## 1. Introduction

Additive manufacturing (AM), commonly known as 3D printing, is a process of joining materials to make objects from 3D computer aided design (CAD) data, usually layer upon layer, as opposed to subtractive manufacturing methodologies [1,2,3]. Powder bed fusion processes, like laser-based powder bed fusion (selective laser melting (SLM)), electron beam-based powder bed fusion processes or electron beam melting (EBM), have gained increasing attention for the fabrication of metallic components [4,5]. EBM uses a high-energy electron beam to selectively melt a conductive metal powder bed directed by a CAD model under a high vacuum. EBM is capable of producing fully dense, near-net-shape complex parts with exceptional mechanical properties [6,7]. SLM, which emerged in the late 1980s and 1990s, uses a laser beam during the fabrication process for the selective melting of metallic powders [8,9,10]. Both SLM and EBM offer the flexibility to produce parts of any shape (theoretically) without restrictions [11,12,13]. SLM can process a wider spectrum of alloys, whereas EBM can process only limited alloys (e.g., Ti-6Al-4V, pure-Ti, CoCrMo and Ni-based alloys). Some other differences exist between the SLM and EBM processes. The difference in the heat source used means that the focus spot size is different—typically ~80 µm in diameter for SLM and ~100 µm in diameter for EBM [9]. Furthermore, typical particle sizes used are 10–60 µm for SLM and 60–105 µm for EBM processes, respectively [9]. The differences in spot size of the heat source and particle size can lead to differences in the size and shape of the melt pool, which subsequently affect the solidification behavior. Even though SLM and EBM share many common features, the minor differences between these two processes could lead to significant variations in the microstructure and, in turn, their mechanical properties [4,6,9]. Ti-6Al-4V is the most prevalent Ti-based alloy and one of the most important engineering materials. Due to its high strength-to-weight ratio, good biocompatibility and outstanding corrosion resistance, Ti-6Al-4V has been widely used in aerospace, biomedical implants, marine and offshore, etc. [13,14,15,16]. With the development of AM, Ti-6Al-4V alloy has been fabricated using both EBM and SLM techniques and has been extensively studied. Compared to the Ti-6Al-4V alloy produced by traditional forging, the SLM/EBM-fabricated materials show a unique microstructure and properties [17,18,19,20,21]. The wear property of Ti-based alloys is one of the most important properties to be considered for particular applications [22,23,24,25,26]. The wear properties are not, however, extensively studied and compared. It is important to know the variation in wear properties of the Ti-6Al-4V alloy produced by both SLM and EBM processes, which means differences in their microstructure. Hence, this work aims to investigate the wear properties of Ti-6Al-4V alloy produced by forging, SLM and EBM. The differences in the wear properties are studied and a possible mechanism for the correlating different microstructures is explained. 

## 2. Experimental Details

Conventional forged samples of Ti-6Al-4V were used to compare the properties with the SLM and EBM samples. The condition of the forged sample is designated as forging a rod with a diameter of 520 mm, hot rolled to 200 mm × 200 mm long squares, subsequently forged to 60 mm × 60 mm long squares, and finally rolled into 14 straight strips. The rolled samples were then annealed at a temperature of 700~800 °C for 1~3 h, and cooled in the air so as the obtain the best possible properties out of the Ti-6Al-4V alloy. The SLM samples were fabricated using a Realizer 50 device with the standard parameter set (laser power—25 W, spot size—~100 µm, laser scan speed—200 mm/s and layer thickness—50 µm). The samples were processed in an argon atmosphere. The powder used for the SLM samples was supplied by Realizer 50 (Realizer, Borchen, North Rhine-Westphalia, Germany) with an average diameter of 30 µm. An Arcam A2X device (ARCAM, Mölndal, Västra Götaland, Sweden) was used to fabricate the sample using the EBM process. Conventional EBM process parameters included: vacuum—10^−4^–10^−5^ (mbar), accelerating voltage—60 kV, layer thickness—50 µm, scan speed—0.50 m/s and the process chamber was maintained at 973 K (the powder bed was pre-heated to 973 K before melting each layer). A core shell scan strategy (with 0.5 mm contour) was used with the melting sequence varying between 0° and 90° between layers [10] for both SLM and EBM samples. The powder used for the EBM samples was supplied by Arcam with an average diameter of 65 µm.

Friction and wear tests were performed on a friction wear testing machine (MM-2000) produced by Jinan Sida Test Technology Company Limited in Jinan, China. The test configuration is shown in Figure 1 below. The wear tests were performed using a ring made of GCr15 alloy with a diameter of 47.15 mm and a thickness of 10 mm, under an applied load of 50 N for a distance of 592.5 m in an air environment with relative humidity ranging from 50% to 60% and at ambient temperature (about 25 °C). Four friction and wear tests were performed for each material type. The GCr15 alloy refers to a high carbon chromium-bearing steel, which has a chemical composition (unit wt. %) as follows: C 0.95–1.05, Mn 0.20–0.40, Si 0.15–0.35, S ≤ 0.020, P ≤ 0.027, Cr 1.30–1.65, Mo ≤ 0.10, Ni ≤ 0.30, Cu ≤ 0.25, Ni + Cu ≤ 0.50, the balance is Fe. This alloy has a tensile strength of 861.3 MPa, yield strength of 518.4 MPa, elongation of 27.95%, bending strength of 1821.6 MPa and hardness of ~630 Hv. The wear test samples were cut to 8 mm × 8 mm × 8 mm by wire cutting, and then they were polished down to 1.5 µm using SiC abrasive paper (wet) and ultrasonically cleaned with ethanol. The sample surface was flat. The samples were weighed to an accuracy of 0.0001 g. The tribological test direction was vertical to the building and forging direction. A new wheel was used for each test. Before and after the test, the samples were cleaned ultrasonically in an alcohol bath for 3 min and dried by air blower. A computerized system was used to record the test parameters such as frequency, load, duration, friction coefficient and speed. The frequency is the number of rotations in one minute of the wheel rotating. The wear rate (WR) was evaluated by the following equation [27,28]:WR = *V_s_*/*L_s_*(1)
where *L_s_* is the sliding distance and *V_s_* is the sliding volume loss. The volume loss was calculated from the wear loss determined by measuring the weight of samples before and after tests. The sliding distance is given by $L_{S}=\pi d_{s}v_{s}t_{s}$, where $d_{s}$ is the diameter of the ring, $v_{s}$ is speed (400 rpm), and $t_{s}$ is the time (10 min) [27,28]. The wear scar was measured according to the depth and width of the longitudinal midsplit face. The microstructure and wear tracks were studied by optical microscopy (OM) using DMI5000 device as well as scanning electron microscopy (SEM) using NOVA NANSEM 430 equipped with an energy dispersive spectroscopy (EDS) analysis. Phase analysis was done by X-ray diffraction (XRD) using a Bruker D8 ADVANCE X-ray diffractometer fitted with Cu-kα radiation (step size—0.01), which was performed on the surfaces vertical to the forging or building directions. The Vickers microhardness was performed using a Vickers microhardness tester from China with a load of 100 N and a dwelling time of 15 s.

## 3. Results

### 3.1. Microstructural Observation

The structural characterization of the Ti-6Al-4V forged, SLM and EBM samples is shown in Figure 2. The XRD pattern shows the presence of a hexagonally close-packed (hcp) phase in all the samples produced by forging, SLM and EBM processes. Here, diffraction peaks corresponding to hcp Ti (JCPDS Card No. 89-3725) were used for comparison, which has lattice parameters a = 0.294 nm and c = 0.467 nm, and shows the standard diffraction peaks for hcp Ti (α phase), located at 2θ = 38.446° and 2θ = 40.177°. The XRD diffraction patterns of the SLM sample indicate the presence of a hexagonal phase with diffraction peaks at 2θ = 38.581° and 2θ = 40.478°, which has lattice parameters a = 0.292 nm and c = 0.467 nm. These values correspond more to the lattice parameters for the α′ phase, i.e., a = 0.2931 nm and c = 0.4681 nm reported in the Materials Properties Handbook by Boyer and Collings (1994) [29]. On the other hand, the EBM sample shows a hexagonal phase with lattice parameters a = 0.299 nm and c = 0.476 nm, indicating the presence of a supersaturation α phase. The matensitic α′ phase in the SLM sample and α phase in the EBM sample are confirmed by the SEM micrograph, where the EBM sample shows a singular shape phase and the SLM sample shows a finer acicular shape phase, which are identified as the α phase and α′ martensitic phase, respectively. These findings are in accordance with the reported work [5,21,25,30,31], revealing that the α′ martensitic phase is typical for SLM-processed Ti-6Al-4V samples. The SLM process is said to offer a very high cooling rate in the order of 10^5^–10^6^ °C/sec [32,33,34]. Such high cooling rates are not, however, observed during conventional forging/EBM processes, hence the difference in the microstructure. 

Furthermore, the XRD patterns seen in Figure 2 reveal that the SLM sample shows high intensities at (10-10) and (10-11) planes, indicating the presence of texture of (10-10) and (10-11) planes in the SLM samples. The EBM sample shows high intensities at the (10-10) plane, indicating the presence of texture of the (10-10) plane. In contrast, the forging sample shows a relatively random distribution of planes. XRD patterns of both forged and EBM samples display peaks of α and β phases. However, the peaks of the β phase are not observed in the SLM samples, suggesting the absence of a β phase or, alternatively, the possible presence of β phase in low concentrations (<5%), which makes it difficult to deduct by XRD [28]. 

It is worth noting that some XRD peaks of the SLM and EBM samples are slightly shifted from the originally expected 2θ positions, and broadened compared to the forged sample. Among the three XRD patterns, the SLM pattern shows the widest peaks, which may be due to the presence of nano-crystalline phases obtained from high cooling rates during solidification or due to the presence of a high degree of internal stresses [10]. 

The OM and SEM images of Ti-6Al-4V alloy manufactured by forging, SLM and EBM processes are shown in Figure 3. Both the SLM and EBM samples have long columnar prior β grains growing along the building direction, but it shows more equiaxed grain in the forged sample, as seen in Figure 3a,b,e,f,i,j. Due to the different solidification conditions, the SLM sample showed acicular α′ martensite filling the columnar prior β grains, while the EBM sample showed singular α phase filling the columnar prior β grains, which is in accordance with the published work [35,36,37]. Black lines are clearly visible inside the columnar prior β grains (as seen in Figure 3e,i), which were identified as the interface between the acicular α′ martensitic and singular α phases for the SLM and EBM samples, respectively. The average width of the columnar prior β grains was ~106 ± 11 µm and ~162 ± 25 µm for the SLM and EBM samples, respectively, indicating that the SLM sample had smaller columnar grains than the EBM sample. The width of the acicular α′ martensite in the SLM sample was found to be 1.1 ± 0.4 µm and the singular α bulge in the EBM sample was found to be 1.4 ± 0.3 µm. Meanwhile, the forged sample exhibited equiaxed α grains with irregular β phase distributed in the α grains or at the boundaries. 

### 3.2. Friction and Wear Properties

The XRD analysis of the tribo-layer is shown in Figure 2b. It shows that the diffraction peaks correspond more to the hcp α phase rather than the martensite α′ phase, indicating that the martensite phase underwent transformation to the α phase during the test. Meanwhile, it reveals that the texture of (10-10) and (10-11) in the SLM sample and the texture of the (10-10) plane in the EBM sample disappeared or significantly weakened after testing, while the peak of (0002) plane for all the samples became much stronger, showing that the texture of the (0002) plane occurred during tribological testing. In addition, all the peaks became wider, indicating added residual strain in these samples.

The coefficient of friction (COF) as a function of sliding distance for Ti-6Al-4V samples is shown in Figure 4. The SLM sample showed the lowest COF, which is 0.4 ± 0.2, the EBM sample had a COF of 0.5 ± 0.1 and the forged sample had the highest COF of 0.6 ± 0.3. The higher the COF, the worse the wear of the property will be. Similarly, in the present case, the forged sample exhibited the highest COF and, hence, the worst wear property amongst the forged, SLM and EBM samples. The COF was determined in relation to the alloying element and composition of the alloy. This caused a difference in the microstructure, hardness and strength in the alloy, thereby affecting its COF [30,38,39]. 

The hardness, weight loss and wear rate for the forging, SLM- and EBM-fabricated Ti-6Al-4V samples is shown in Figure 5. Microstructural differences lead to different hardness levels (as seen in Figure 5a) in these samples, where the highest hardness of 399 ± 14 HV was observed for the SLM sample as compared to the EBM (383 ± 13 HV) and forged (368 ± 12 HV) samples. SEM micrographs of the samples after Vickers indentation are shown in Figure 6, which shows that the smallest indentation size obtained in the SLM sample indicated the highest hardness compared to the other samples by forging and EBM processes. Finer microstructure (fine needle martensite) of samples produced by additive manufacturing had higher hardness than the forged sample. As expected from the hardness plot, the forged sample showed the highest wear rate (23.9 ± 4.6 × 10^−5^ mm^3^ N^−1^ m^−1^). The EBM sample, however, showed the lowest wear rate (16.6 ± 4.2 × 10^−5^ mm^3^ N^−1^ m^−1^) as compared to the SLM and the forged sample. The SLM sample showed a wear rate of 19.0 ± 3.7 × 10^−5^ mm^3^ N^−1^ m^−1^, proving that the EBM sample showed the highest wear resistance among the three considered samples. 

SEM images of the worn surfaces can be seen in Figure 7. All of the Ti-6Al-4V samples fabricated by forging, SLM and EBM processes showed the presence of typical wear scars and ploughing grooves along the sliding direction. The wear scars observed on the forged samples (depth of 0.24 ± 0.01 mm) were deeper than the SLM (depth of 0.21 ± 0.03 mm) and EBM (depth of 0.17 ± 0.02 mm) samples. The EBM sample showed the shallowest wear scar. Hard particles, such as metallic debris and oxide particles, formed and were reduced to a fine debris under continuous sliding, which can lead to ploughing grooves as observed in the tribo-layer (Figure 7a,d,g). The wear test with high strain raised the temperature of the surface in contact with the counter disc. Subsequently, the debris was compacted on the worn surfaces to generate the tribo-layer, as seen in Figure 8 and Figure 9. The EDX results are shown in Table 1. 

High oxygen content was observed in the tribo-layer, suggesting that the oxidation occurred during the sliding test. Meanwhile, a high Fe content presented in the tribo-layer, which was mostly from the steel wheel during testing. The tribo-layer showed high carbon content as seen in Table 1. In addition, there were higher carbon content presences beneath the tribo-layer as seen in Figure 9. The high carbon content observed in and beneath the tribo-layer may have formed during the wear testing and/or may have come from the original sample surface that was exposed to the laboratory environment and handling. A much deeper and systematic study is, however, needed to address this issue. A continuous deep horizontal crack was observed at the bottom of the tribo-layer in the forged sample (as seen in Figure 8), resulting in high wear rates due to the delamination initiated by these sub-surface cracks. More horizontal cracks were also observed in the SLM sample but these were discontinuous in nature. On the other hand, the EBM sample showed few horizontal cracks but many small vertical cracks. 

## 4. Discussion

Based on the above results, it can be observed that the SLM and EBM Ti-6Al-4V samples showed predominantly similar wear mechanisms. An abrasive wear due to the loading from the GCr15 ring and tribo-layer was due to plastic deformation and oxidation. The SLM and EBM samples showed better wear resistance than the forged sample. This result can be correlated to the higher hardness and weaker delamination tendency in the sub-surface of the SLM and EBM samples during the wear tests. Moreover, the SLM and EBM samples exhibited higher hardness than the forged sample, which can be ascribed to the presence of fine microstructure and martensitic phase (as seen in Figure 3). Severe delamination of the tribo-layer was developed in the forged sample due to the formation of the primary crack during the cycling loading, leading to significant material removal (as seen in Figure 7b), which is in accordance with other reported works [40,41,42,43]. Unlike the forged sample, the SLM and EBM samples showed more uniformly distributed cracks, which can avoid the formation of primary cracks. 

Although the EBM samples had lower hardness in comparison to the SLM samples, they showed a lower wear rate (as seen in Figure 5). This may be mainly ascribed to the formation of several horizontal cracks in the SLM sample resulting in the relatively severe delamination of the tribo-layer—in comparison to the EBM sample, which had more vertical cracks and only a few horizontal cracks. After the delamination of the tribo-layer, a new tribo-layer was formed at the nascent surface of the Ti-6Al-4V SLM sample during continuous loading, leading to relatively severe mass loss in the SLM sample in comparison to the EBM sample. 

A schematic illustration of the different wear behavior in Ti-6Al-4V produced by the different processes is shown in Figure 10. The tribo-layer and the cracks were highlighted in the figures, which are the most important facts affecting the wear resistance of the forged, SLM and EBM samples. Most of the cracks propagated horizontally in the SLM samples, which could lead to delamination during the continuous wear loading. Meanwhile, most of the cracks propagated vertically in the EBM samples, giving them better resistance than the SLM samples. The main reason for the crack propagation behavior resulted from the different microstructure, in which a more brittle martensitic phase existed in the SLM sample. 

## 5. Conclusions

In this work, Ti-6Al-4V samples were produced by forging, SLM and EBM processes, and their resultant microstructure, hardness and wear behaviors were compared. The microstructural characterization revealed that the forged sample had equiaxed α grains with an irregular β phase distributed in α grains or at their boundaries. The SLM sample showed the presence of an acicular α′ martensite phase, while the EBM sample showed a singular α bulge phase distributed in the columnar prior β grains. The diameter of the columnar prior β grains was smaller in the SLM sample than in the EBM sample. The SLM sample showed the highest hardness owing to the fine microstructure and α′ martensitic phase. The SLM and EBM samples showed better wear resistance than the forged sample, which correlate to their higher hardness and weaker delamination in the AM samples. Furthermore, the EBM sample showed a lower wear rate than the SLM sample owing to the fact that a number of horizontal cracks were formed in the SLM sample, resulting in the heavier delamination of the tribo-layer. 

## Figures and Tables

**Figure 1 materials-12-00782-f001:**
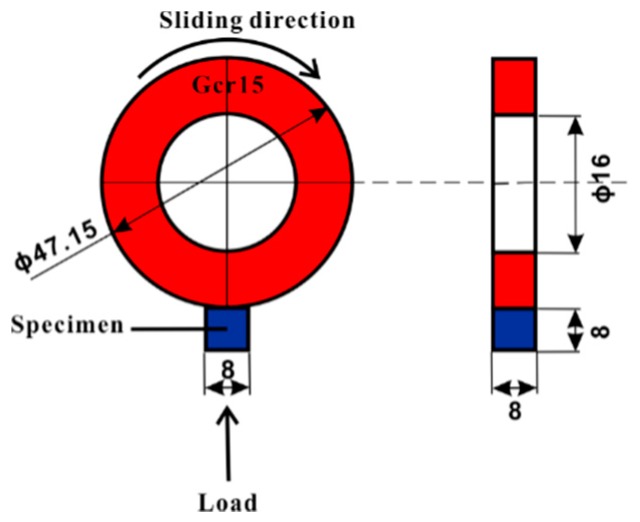
A schematic showing the wear test configuration including the position of the specimen, sliding direction and dimensions of the wheel.

**Figure 2 materials-12-00782-f002:**
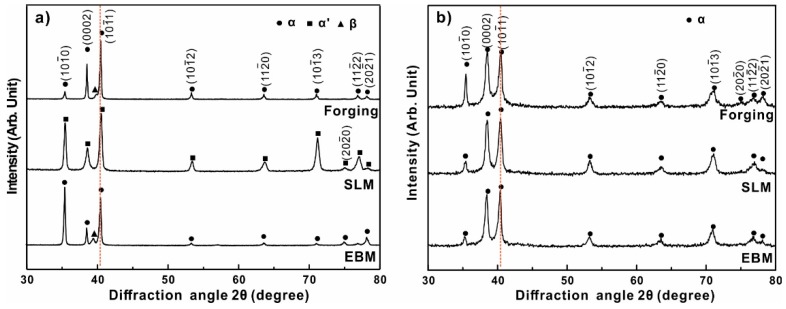
X-ray diffraction (XRD) analysis of Ti-6Al-4V samples produced by forging, selective laser melting (SLM) and electron beam melting (EBM) processes: (**a**) before sliding wear tests, (**b**) the tribo-layer surface after sliding wear tests.

**Figure 3 materials-12-00782-f003:**
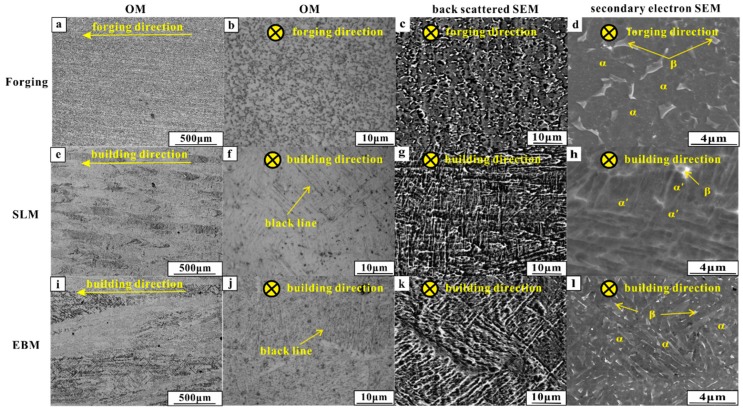
OM, back scattered SEM and secondary electron SEM images of samples produced by (**a**–**d**) forging; (**e**–**h**) SLM; (**i**–**l**) EBM.

**Figure 4 materials-12-00782-f004:**
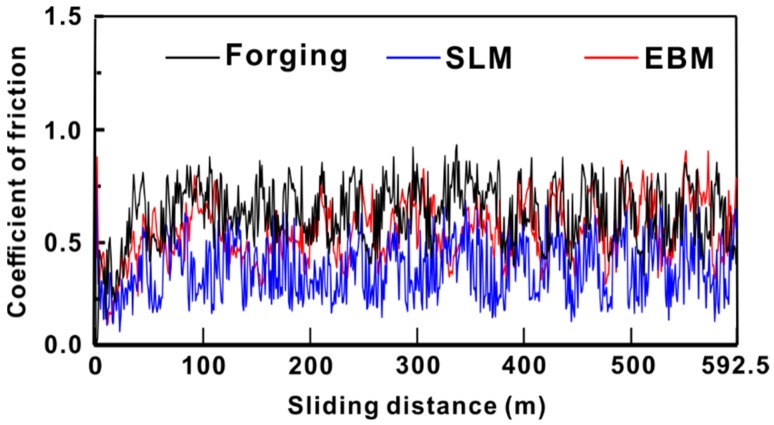
Coefficient of friction values for Ti-6Al-4V samples produced by forging, SLM and EBM.

**Figure 5 materials-12-00782-f005:**
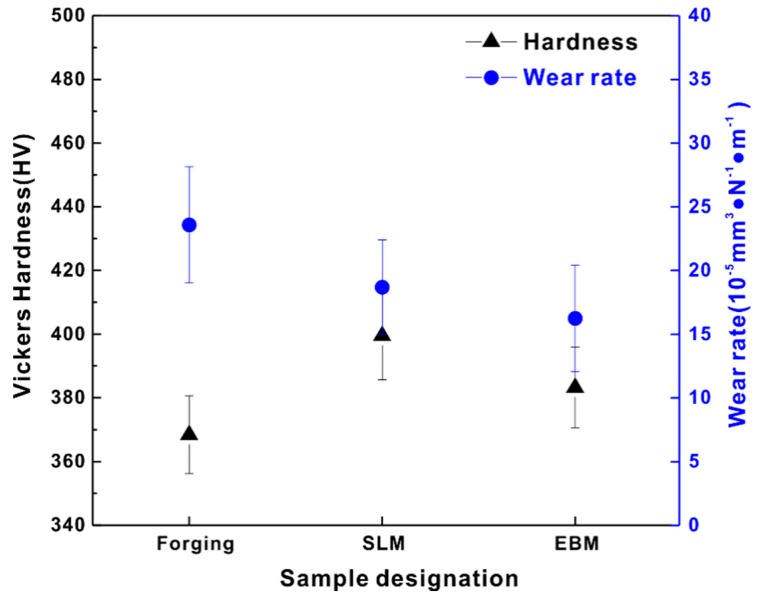
Hardness and wear rate of Ti-6Al-4V samples fabricated by forging, SLM and EBM.

**Figure 6 materials-12-00782-f006:**
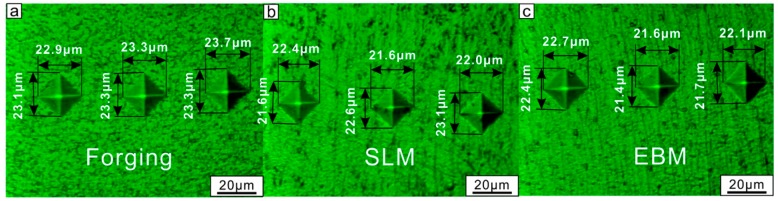
SEM micrographs of the samples after Vickers indentation: (**a**) forged, (**b**) SLM and (**c**) EBM.

**Figure 7 materials-12-00782-f007:**
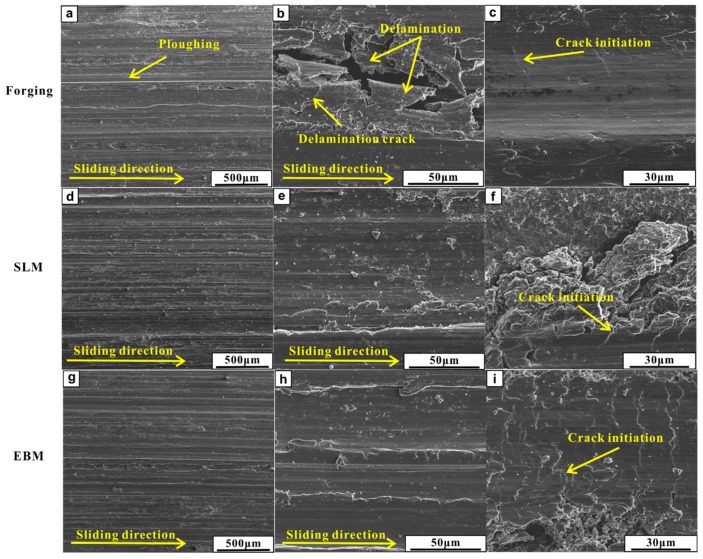
SEM images of wear tracks after the sliding wear tests for the Ti-6Al-4V samples: (**a**–**c**) forged, (**d**–**f**) SLM and (**g**–**i**) EBM.

**Figure 8 materials-12-00782-f008:**
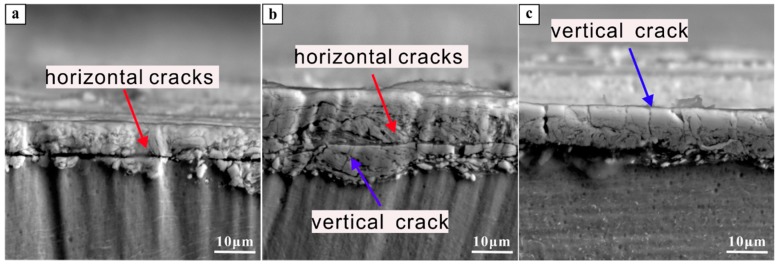
Cross-section comparison of Ti-6Al-4V alloy produced by (**a**) forging, (**b**) SLM and (**c**) EBM.

**Figure 9 materials-12-00782-f009:**
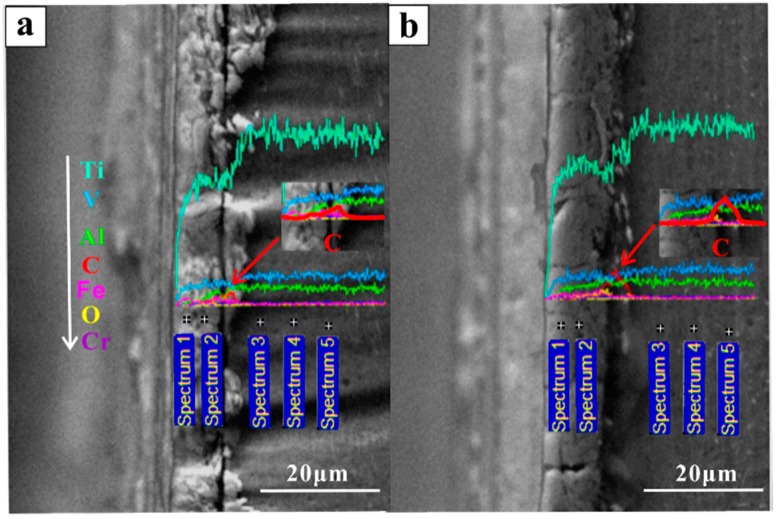
Energy dispersive spectroscopy (EDS) lines and points analysis of tribo-layer of the cross-sections of the (**a**) forged, (**b**) EBM samples.

**Figure 10 materials-12-00782-f010:**
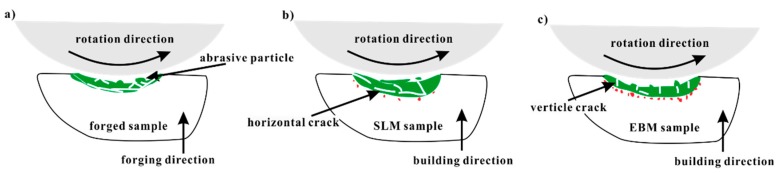
Schematic illustration of the wear mechanisms in Ti-6Al-4V samples produced by (**a**) forging, (**b**) SLM and (**c**) EBM processes.

**Table 1 materials-12-00782-t001:** Element analysis (wt. %) of the tribo-layer and matrix of the samples produced by forging, SLM and EBM processes.

Process	Position	C (%)	O (%)	Al (%)	Ti (%)	V (%)	Fe (%)
Forging	tribo-layer	6.10 ± 3.85	5.73 ± 4.90	2.24 ± 0.85	77.39 ± 7.00	3.45 ± 0.28	4.86 ± 1.92
	matrix	1.30 ± 0.40	−2.26 ± 1.17	5.96 ± 0.19	91.14 ± 1.24	3.72 ± 0.27	0.19 ± 0.10
SLM	tribo-layer	5.91 ± 1.49	10.43 ± 0.99	3.40 ± 0.69	72.37 ± 2.62	2.97 ± 0.40	4.93 ± 0.11
	matrix	0.90 ± 0.31	−1.63 ± 0.40	5.25 ± 0.36	91.90 ± 0.75	3.49 ± 0.13	0.06 ± 0.09
EBM	tribo-layer	2.47 ± 2.14	12.37 ± 5.10	3.29 ± 0.34	75.08 ± 6.80	3.02 ± 0.37	3.77 ± 0.46
	matrix	1.27 ± 0.35	−1.68 ± 1.27	5.60 ± 0.24	91.51 ± 0.96	3.18 ± 0.43	0.07 ± 0.01

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
