# Peer review of "Superior Wear Resistance in EBM-Processed TC4 Alloy Compared with SLM and Forged Samples"

_materials, 2019, doi:10.3390/ma12050782_

Round 1
Reviewer 1 Report
1. About tribological tests: what's about humidity? Was it controlled? What about contact temperature, was it measured? How was chosen the angular velocity of 400 rpm and the load 50N?
2. What with sliding distance: was 188.6m (page, 3 line 81) or 592.5 m(Fig. 4, page 5 line 135)
3. -How many friction and wear tests were performed for each material type?
4. What happens with the microstructure of the samples during tribological tests? Whether martensite phase undergoes transformation during the test, or disintegration? Was the contact temperature (counter sample – specimen) measured?
5. The tribological tests were done under a load 50N and fixed velocity. A much more comprehensive experimental campaign should be conducted at different loads and velocities to characterize the wear and friction behaviors. The current set of data are not sufficient to publish
6. Page 5, line 152: Authors write, than: “The wear scars observed on the forged samples are deeper than SLM and EBM samples….” this cannot be statement on the basis of images obtained by SEM microscope – more detailed investigations should be done, e.g. confocal microscopy can be used for measurements
7. Page 5 line 155 – text written in a different character
8. Page 6 line 165 - Authors write: “are rich in carbon (Fig. 8c) , indicating the possible formation of carbides during the wear experiments– there is no Fig. 8c in the article…. whether such wear process conditions are favorable for the carbides formation? if so, please explain it further basis on the theory and results.
9. Page 7 line 194, Authors write: “Carbide may be formed below the tribolayer and may improve the wear resistance since it has much higher hardness than Ti alloys. Carbon atom transferring from steel ring (GCr15) to surface of samples during adhesion process under high speed sliding test maybe the reason of formation of carbide phase under local high temperature at the contact area”
- I do not think that carbon diffusion into the sample could occur with such process parameters…
Please explain this theory, which carbides can form in these process conditions, Authors should carry out a thermodynamic analysis of the carbides formation during friction, and mandatory determine the temperature to which the sample heats up during friction.
10. Why Authors didn’t perform the tribological tests in the conditions of fluid friction?
Author Response
We would like to thank the reviewers for their critical comments and constructive suggestions, which are greatly helpful to improve the quality of the manuscript. All their comments have been taken into consideration and revised in our manuscript accordingly in red. We provide below our responses to the reviewers’ comments point by point:
1. About tribological tests: what's about humidity? Was it controlled? What about contact temperature, was it measured? How was chosen the angular velocity of 400 rpm and the load 50N?
Response: The tribological tests were performed in air environment of relative humidity ranging from 50% to 60%, which is not controlled. The tribological tests were performed at ambient temperature of about 25℃. Different angular velocity ranging from 200-500 rpm and different load ranging from 30 N to 60 N were chosen for the tribological tests. It is found that when the angular velocity and load is too low, it is vague to confirm the difference of the wear behavior of the TC4 alloys produced by different processes such as EBM, SLM and forging samples. However, when the angular velocity and load is too high, the size of the wear scars and ploughing grooves are too large which are even larger than the sample width. At last, we found that the parameters with the angular velocity of 400 rpm and the load 50N were found the most suitable ones for comparing the wear behavior.
2. What with sliding distance: was 188.6m (page, 3 line 81) or 592.5 m(Fig. 4, page 5 line 135)
Response: Thanks the reviewer’s carefully read. It is 592.5 m.
3. How many friction and wear tests were performed for each material type?
Response: Four friction and wear tests were performed for each material type, which was included in the revised manuscript.
4. What happens with the microstructure of the samples during tribological tests? Whether martensite phase undergoes transformation during the test, or disintegration? Was the contact temperature (counter sample – specimen) measured?
Response: we performed XRD test for the samples after tribological testing. Fig. 2b shows the XRD analysis of the tribo-layer. It shows that the diffraction peaks are more corresponding to hexagonal close-packed (hcp) α phase rather than martensite α′ phase, indicating that martensite phase undergoes transformation to α phase during the test. Meanwhile, it reveals that the texture of (10-10) and (10-11) in the SLM sample and texture of (10-10) plane in the EBM sample disappeared or significantly weakened after testing, while the peak of (0002) plane for all the samples becomes much stronger showing that the texture of (0002) plane occurred during tribological testing. In addition, all the peaks become wider indicating added residual strain in these samples. The contact temperature was not measured and we will perform it for the works in the near future.
5. The tribological tests were done under a load 50N and fixed velocity. A much more comprehensive experimental campaign should be conducted at different loads and velocities to characterize the wear and friction behaviors. The current set of data are not sufficient to publish
Response: we fully understand the concerns made by the reviewer, a much more comprehensive experimental campaign is needed to characterize the wear and friction behavior. As mentioned above, we chose this fixed parameter because it is not available to get applicable or repeatable results if the load or velocity is too low or too high. For example, the coefficient of friction curve is drastically fluctuated when the load is too low, as shown in the Figure R1 below. However, the purpose is trying to show some of the general wear behavior of the TC4 produced by different processes; with the fixed parameter, it can be seen that TC4 alloy produced by forging, SLM and EBM processes shows different wear behavior by comparing the wear rate, worn surface and so on. In the near future, we will change the friction wear testing machine or even using the fretting wear testing machine to further study the wear and friction behaviors, and more results and discussion will be completed and submitted in other works.
Figure R1 Coefficient of friction values for Ti-6Al-4V samples produced by EBM process with different applied load during wear tests.
6. Page 5, line 152: Authors write, than: “The wear scars observed on the forged samples are deeper than SLM and EBM samples….” this cannot be statement on the basis of images obtained by SEM microscope – more detailed investigations should be done, e.g. confocal microscopy can be used for measurements
Response: we tried to characterize the three-dimensional topography of each wear scar by the BMT Expert3D morphology analyzer, however, the wear scars are too deep to measure. During our revision process, we cut the wear tested samples along the longitudinal midsplit face where the depth and width of the wear scar were measured, as shown in the Figure R2 below.
Figure R2 The photos show the methods to measure the wear scars.
7. Page 5 line 155 – text written in a different character
Response: The text was corrected in the revised manuscript.
8. Page 6 line 165 - Authors write: “are rich in carbon (Fig. 8c) , indicating the possible formation of carbides during the wear experiments– there is no Fig. 8c in the article…. whether such wear process conditions are favorable for the carbides formation? if so, please explain it further basis on the theory and results.
Response: It is Fig. 8b. We agree with the comments raised by all the reviewers that more proof is needed to confirm this conclusion. It needs a much deeper and systematic study, which will be even good enough to address in a separate work. Therefore, we do not emphasize the formation of carbide in the revised manuscript.
9. Page 7 line 194, Authors write: “Carbide may be formed below the tribolayer and may improve the wear resistance since it has much higher hardness than Ti alloys. Carbon atom transferring from steel ring (GCr15) to surface of samples during adhesion process under high speed sliding test maybe the reason of formation of carbide phase under local high temperature at the contact area”
- I do not think that carbon diffusion into the sample could occur with such process parameters…
Please explain this theory, which carbides can form in these process conditions, Authors should carry out a thermodynamic analysis of the carbides formation during friction, and mandatory determine the temperature to which the sample heats up during friction.
Response: we noted that the formation of carbide is not convincible in this work and our expression should be modified to avoid making conclusion in excess of the results. The research on the theory and experiments as mentioned by the reviewer is interesting and helpful to understand the wear behavior of the SLM and EBM fabricated TC4 samples, which will be considered to perform in the near future.
10. Why Authors didn’t perform the tribological tests in the conditions of fluid friction?
Response: The tribological tests were performed on a friction wear testing machine (MM-2000) in our lab, which does not support the conditions of fluid friction.

Reviewer 2 Report
Most of my comments to the authors are contained in the attached PDF. I suggest that the editors suggest that most of the points I raise are addressed.

Author Response
We would like to thank the reviewers for their critical comments and constructive suggestions, which are greatly helpful to improve the quality of the manuscript. All their comments have been taken into consideration and revised in our manuscript accordingly in red. We provide below our responses to the reviewers’ comments point by point:
1. For both SLM and EBM please provide details of the powder used to manufacture the samples.
Response: As suggested by the reviewer, necessary information regarding the powder used were included in the manuscript.
2. Please describe materials that were produced more fully. For example, are they all fully dense (was this measured)? Were they subsequently machined or heat treated in any way? HOW?
Response: The best parameter combinations were used to fabricate both SLM and EBM samples. The density of the samples were measured using Archemedes principle, where the relative density was found to be ~99.5% or more. No further machining nor heat treatment were performed on these samples.
3. What is a core shell strategy? Explain and compare with the strategy used in the laser situation.
Also, how thick were the SLM and EBM layers and how many passes were used to create them?
Response: The necessary information were introduced.
4. Manufacturer ? ( friction wear testing machine)
Response: The manufacturer of friction wear testing machine is produced from Jinan sida Test technology co. Ltd in Jinan of China.
5. what is this(GCr15)? more details and properties, Was it hardened? How hard?
Response: The GCr15 alloy refers to a high carbon chromium bearing steel, which has a chemical composition (unit wt.%): C 0.95-1.05, Mn 0.20-0.40, Si 0.15-0.35, S≤0.020, P≤0.027, Cr 1.30-1.65, Mo≤0.10, Ni≤0.30, Cu≤0.25, Ni+Cu≤0.50, the balance is Fe; and it has a tensile strength of 861.3 MPa, yield strength of 518.4 MPa, elongation of 27.95%, bending strength of 1821.6 MPa and hardness of ~630 Hv. It is much harder than the TC4 alloy produced by forging (368 ± 12 Hv), SLM (399 ± 14 Hv) and EBM (383 ± 13 Hv) processes in this work.
6. Also describe the samples, size, shape, surface finish. Was the sample surface flat or curved to match the curvature of the wheel?
Response: The size, shape and surface finish were added in the experimental section. The sample surface was flat. The wear test samples were cut to 8mm×8mm×8mm by wire cutting, and then they were polished down to 1.5 µm SiC abrasive paper (wet) and ultrasonically cleaned with ethanol. The specimens were weighed to an accuracy of 0.0001 g.
The manuscript was revised accordingly.
7. What was test direction compared to build or forging direction?
Response: The tribological test direction is verticle to the building and forging direction.
The manuscript was revised accordingly.
8. Also, how many samples of each type were tested (presumably more than one sine there are error bars in the results).
Response: Four samples of each type were tested.
9. Was a new wheel used each time or was it cleaned or re-prepared before each test?
What order were the samples tested in?
Response: we used new wheel for each sample.
The manuscript was revised accordingly.
10. 592.5m according to later graphs and data below
Response: Yes, it is 592.5 m. The mistake was corrected in the revised manuscript.
11. frequency of what?
Response: it is rotation number in one minute of the wheel rotated.
12. missing comma? do you mean load, duration?(Line 83: load duration)
Response: Thanks the reviewer’s carefully read. It missed a comma, which we mean load and duration.
The manuscript was revised accordingly.
13. why not just use WR? ( )
Response: we replace Qs with WR in the revised manuscript.
14. give details of sample cleaning procedure before and after the test along with details of the balance used (e.g. 0.001g accuracy)
Response: The details were added in the revised manuscript. Before and after the test, the samples were cleaned ultrasonically in alcohol bath for 3 minutes and dry by air blower, respectively. The weight loss was measured having a precision of 0.0001g.
15. What surface was hardness tested? What was surface finish? Can an image of the 3 hardness tests be shown in results please?
Response: as suggested by the reviewer, one new figure (Figure 6) was supplied in the revised manuscript to show how was hardness tested.
16. How was XRD done? On what surface and which direction with regards the growth direction or forging direction? Give XRD acquisition condtions.
Response: XRD was performed on the surfaces which are vertical to the forging or building directions.
17. please describe in more detail how you have differentiated between alpha and alpha prime microstructures using the XRD data.
Response: According to the comment, we significantly revised the manuscript, which is shown below.
The XRD pattern shows the presence of hexagonally closed packed (hcp) phase in all the samples produced by forging, SLM and EBM processes. Here diffraction peaks corresponding to hexagonal close-packed (hcp) Ti (JCPDS Card No. 89-3725) were used for comparison, which has lattice parameters a = 0.294 nm and c = 0.467 nm, and shows the standard diffraction peaks for hcp Ti (α phase), located at 2θ = 38.446o and 2θ = 40.177o. The XRD diffraction patterns of SLM sample indicate the presence of a hexagonal phase with diffraction peaks at 2θ = 38.581o and 2θ = 40.478o, and has lattice parameters a = 0.292 nm and c = 0.467 nm. These values are more correspond to the lattice parameters for the α′ phase, i.e. a = 0.2931 nm and c = 0.4681 nm reported in Materials Properties Handbook by Boyer and Collings (1994) [1]. While the EBM sample shows a hexagonal phase with lattice parameters a = 0.299 nm and c = 0.476 nm indicate the presence of supersaturation α phase. The matensitic α′ phase in SLM sample and α phase in EBM sample are confirmed by the SEM micrograph, where the EBM sample shows singular shape phase and SLM sample shows finer acicular shape phase, which is identified as α phase and α′ martensitic phase, respectively. These are in accordance with the reported work[2–6].
8. Note also the difference in the intensities of the peaks between sample types. This may suggest preffered orientation of some crystal planes.
Response: the SLM sample shows high intensities at (10-10) and (10-11) planes, indicating the presence of texture of (10-10) and (10-11) planes existed in the SLM samples. The EBM sample shows high intensities at (10-10) plane, indicating the presence of texture of (10-10) plane. While the forging sample shows a relatively random distribution of planes.
19. whats is the forging direction of the forged samples?
Response: The forging direction of the forged samples was provide in Figure 3 in the revised manuscript.
20. please provide a standard deviation or other varaiance statistics for this data and say how the average was arrived at (all data or only steady state data after xx m of sliding?) (line 127: The SLM sample show the lowest COF, which is ~0.4, the EBM sample has COF of ~0.5 and the forged sample has the highest COF ~0.6. Higher the COF worse will be the wear property.)
Response: COF data was obtained from only steady state data after 40 m of sliding, and the standard deviation was added in the revised manuscript.
21. It is not necessarily true to say that higher hardness causes lower COF. There is no evidence here that higher hardness is causal, only that it correlates.
Response: Thanks for the reviewer’s suggestion; it was corrected according to the comment.
22. in the figure (coefficient of friction) use a scale which better shows the data and its variation, friction rarely exceeds 1 and can never be negative.
Response: the figure was redrawn by changing the scale. The data exceeding 1 or negative is noise.
23. in this case weight loss and wear rate are the same data, no need to show both. Also, what was the density of all 3 materials? If the SLM or EBM materials were not sully dense then the wear volume and thus wear rate need to be adjusted accordingly.
Response: the weight loss was deleted in figure 5. The relative density of the SLM and EBM sample is 99.7% and 99.5% respectively. The wear rate was then adjusted accordingly.
The manuscript was revised accordingly.
24. Hardness is not necessarily a predictor of wear rate.
Response: it was avoided in the revised manuscript.
25. it doesn't suggest it, it proves it.( EBM sample shows the lowest (16.6 ± 4.2x10-5 mm3•N-1•m-1)suggesting that EBM sample shows the highest wear resistance among the three considered samples)
Response: thanks the reviewer’s comment, we had modified it accordingly in the text.
26. How was this measured and quantified?(The wear scars observed on the forged samples are deeper than the SLM and EBM samples, which SLM samples shows the shallowest wear scar.)
Response: we cut the wear tested samples along the longitudinal midsplit face where the depth and width of the wear scar were measured, as shown in Figure R2.
Figure R2 The photos show the methods to measure the wear scars.
27. Why does this suggest "homogeneous wear" and in fact what does this even mean?(line 154: Cracks are observed in all the three samples, indicating homogeneous wear taking place in all the three samples during the wear tests)
Response: we delete these words to avoid misunderstanding.
28. Not necessary oxidation of the surface, Oxidation of fine wear debris which are then compacted into a tribo-layer.(line 160: high oxygen content is observed in the tribo-layer suggesting the oxidation of the surface)
Response: the expression was revised according to the comment.
29. Perhaps not a crack but simply the interface between the sample surface and the tribolayer.(line 160: A continue deep horizontal crack is observed at the bottom of the tribo-layer in the forged sample (Fig. 7))
Response: we are sure that they are real cracks. Below the higher resolution image clearly reveals the crack.
Figure R3 Cross-section comparison of Ti-6Al-4V produced by forging process
30. It is equally possible that the carbon signal is coming from the original sample surface beneath the tribolayer as any surface exposed to the normal laboratory environment and handled, even if cleaned with alcohol, will show a significant carbon presence.(line 164: Several tiny particles are observed below the tribo-layer of the SLM and EBM samples, which are rich of carbon (Fig. 8c), indicating the possible formation of carbides during the wear experiments. )
Response: according to this comment, both possible reasons for the carbon signal were discussed in the revised manuscript.
31. How were the cross-sections prepared and where on the wear surface were they taken?
Response: we cut the wear tested samples along the longitudinal midsplit face (shown in Figure R2), then the longitudinal midsplit surface was polished to do SEM observation.
32. Above it is noted that the surface layer showed a high level of oxygen to be present but it is not shown here. Why? I would expect it to be as this is a tribolayer.
There also appears to be a suggestion of Fe being present in the tribolayer but the data is not presented in a way which makes this possible to see. The authors should include a semi-quant EDX spectrum from the tribolayer on all 3 samples.(Fig.8:EDS line)
Response: the line showing the oxygen level in Figure is not clear to see. We provide the point element analysis in the revised manuscript, as shown in Table 1.
33. refer to the original sample identifiers e.g. SLM and EBM (AM)
Response: we changed the sample identifies to SLM and EBM.
34. I would question whether any of this is proven. Yes, there is higher hardness but it is not shown to be causal, only that it correlates, there is no evidence of a hard carbide phase, only of some extra carbon which could just be prior surface contamination. It is not explained in what way the delamination is reduced, how has it been determined that there is less delamination?(line 178: owing to the higher hardness, weaker delamination tendency and formation of hard carbide phase in the sub-surface of the AM samples during the wear tests.)
Response: We oblige with the comments from the reviewer. Accordingly, the formation of carbide needs more detailed proof to support this conclusion. Delamination was clearly observed in some regions of the wear scar of the forging sample, however, it is not observed in the SLM and EBM samples.
35. what is the evidence for "severe delamination"? There are no images showing large missing area of delaminated tribofilm.(line 181: Severe delamination of the tribo-layer was developed in the forged sample )
Response: The evidence for "severe delamination" was displayed in cross-section of wear sample (Fig. 8(a)).
36. The authors need to better define the cracks they have observed and refer to any literature that corroborates their conclusions. They have seen a horizontal crack (which may not actually be a crack) and lots of vertical cracks through the tribolayer. How do these relate to these conclusions? (line 183: the formation of the primary crack during the cycling loading, leading to significant material removal (Fig. 6b). Unlike the forged sample, the SLM and EBM samples show more uniformly distributed cracks, which can avoid the formation of the primary cracks.)
Response: The surface and sub-surface cracks were observed. However, we deal with the primary cracks on the surface, which is because of the plastic deformation and tearing of the surface layer. Excessing deformation leads to the formation of cracks, which is indeed correlated to the phases present in the sample and their hardness. Hence, we are focusing much into the microstructure of the samples produced by the three different manufacturing techniques (forging, SLM and EBM).
37. Which disproves the previous assertions that higher hardness means lower wear rate.(line 186: Although the EBM samples have lower hardness than the SLM samples, it shows a lower wear rate (Fig. 5). )
Response: it was corrected according to the comments.
38. If only small holes are opening in the tribolayer and the rest of the tribolayer is maintained then how is contact made between the steel wheel and the original Ti sample surface? (line 189: After the delamination of the tribo-layer, a new tribo-layer will be formed at the nascent surface of the Ti-6Al-4V SLM sample during continuous loading, leading to relatively severe mass loss in the SLM sample than the EBM sample.)
Response: initially the original Ti sample surface started to form a tribo-layer during testing, then the delamination of the tribo-layer occurred and the fresh (or covered with less tribo-layer) Ti matrix will show up and keep in contact with the wheel due to the 50 N load, which will subsequently form new tribo-layer.
39. very little proof of this provided and it would be very suprising.(line 194: Carbide may be formed below the tribo-layer and may improve the wear resistance since it has much higher hardness than Ti alloys. )
Response: as mentioned above, this claim is not shown in the revised manuscript due to the very little proof, and we will try to address it in the future work.
40. not a viable explanation.(line 197: maybe the reason of formation of carbaide phase)
Response: this explanation is not used in the revised manuscript.
41. Overall this discussion is inadequate in at times inaccurate. Cracks are occuring in the tribofilm, not in the original material, thus the differences in the as manufactured microstructures are irrelevant as the main body of the sample is not involved in this process. The authors instead need to concentrate on how the tribofilm is created and why they might be different for chemically similar alloys. (line 201: The main reason for the crack propagation behavior is resulted from the different microstructure, in which more brittle martensitic phase exists in the SLM sample.)
Response: I agree with the reviewers comments on the discussion about the formation of tribofilm in the wear surface. This is indeed another study (which needs detailed characterization along the interface), which is being carried out presently and will be published as a new manuscript. However, in the context on the present manuscript, we have given the differences in the microstructural features as the main reason for difference in their tribological properties.
42. This diagram misrepresents the real situation where the wheel (47mm) is much bigger than the sample (8mm) (Fig.9)
Response: it was revised in the revised manuscript.
43. this conclusion is not supported by the data. Only a correlation with hardness has been observed, it is not necessarily the cause. There is no real evidence of carbides. Delamination and its extent and measurement have not been defined or quantified. (line 214: The SLM and EBM samples show better wear resistance than the forged sample, owing to the higher hardness, weaker delamination and formation of hard carbide phase in the addictive manufactured samples.)
Response: this conclusion was revised according to the comments contributed by all the reviewers.
References:
1. G.Welsch,R.Boyer,E.W.Collings,MaterilasPropertiesHandbook:Titanium Alloys, ASMInternational,MetalsPark,OH,1994.
2. Tang, B.; Wu, P.Q.; Fan, A.L.; Qin, L.; Hu, H.J.; Celis, J.P. Improvement of corrosion-wear resistance of Ti-6Al-4V alloy by plasma Mo-N surface modification. Adv. Eng. Mater. 2005, 7, 232–238.
3. Vastola, G.; Zhang, G.; Pei, Q.X.; Zhang, Y.W. Active Control of Microstructure in Powder-Bed Fusion Additive Manufacturing of Ti6Al4V. Adv. Eng. Mater. 2017, 19, 1–6.
4. Van Hooreweder, B.; Moens, D.; Boonen, R.; Kruth, J.P.; Sas, P. Analysis of fracture toughness and crack propagation of Ti6Al4V produced by selective laser melting. Adv. Eng. Mater. 2012, 14, 92–97.
5. Balla, V.K.; Soderlind, J.; Bose, S.; Bandyopadhyay, A. Microstructure, mechanical and wear properties of laser surface melted Ti6Al4V alloy. J. Mech. Behav. Biomed. Mater. 2014, 32, 335–344.
6. Gu, D.; Hagedorn, Y.; Meiners, W.; Meng, G. Densification behavior , microstructure evolution , and wear performance of selective laser melting processed commercially pure titanium. 2012, 60, 3849–3860.

Reviewer 3 Report
The paper entitled "Superior wear resistance in the EBM-processed TC4 alloy compared with the SLM and forged samples" presents the applicability of EBM for TC4 alloy as a wear–resistant alloy. I think the paper is interesting because it opens new perspectives in the use of additive manufacturing process. However, there are some points to be improved as following:
1. You should mention the preheating process in EBM as the difference between SLM and EBM in the Introduction section.
2. You should explain the difference of the intensity ratio of the peaks originated from α–phase in the forging and EBM in Fig.2.
3. What do "Black lines" at 118 lines mean ? You should need to explain further.
4. What does the under line of "cyclic load" at 155 lines mean ?
5. I can not find Fig.7 (f, i) at 156 lines.
6. What is "tribo-layer" at 159 lines ? You should need to explain further.
7. I can not find Fig.8c at 165 lines.
8. What does the arrow showed in Fig.8a mean ?
9. Fig.9 does not explain any wear mechanisms. For example, you should explain the effects of the carbides originated from the steel ring and AM process by using Fig.9.
Author Response
We would like to thank the reviewers for their critical comments and constructive suggestions, which are greatly helpful to improve the quality of the manuscript. All their comments have been taken into consideration and revised in our manuscript accordingly in red. We provide below our responses to the reviewers’ comments point by point:
The paper entitled "Superior wear resistance in the EBM-processed TC4 alloy compared with the SLM and forged samples" presents the applicability of EBM for TC4 alloy as a wear–resistant alloy. I think the paper is interesting because it opens new perspectives in the use of additive manufacturing process. However, there are some points to be improved as following:
1. You should mention the preheating process in EBM as the difference between SLM and EBM in the Introduction section.
Response: as suggested by the reviewer, the preheating process is mentioned in the revised manuscript.
2. You should explain the difference of the intensity ratio of the peaks originated from α–phase in the forging and EBM in Fig.2.
Response: the difference of the intensity ratio of the peaks was explained in the revised manuscript. It reveals that the SLM sample shows high intensities at (10-10) and (10-11) planes, indicating the presence of texture of (10-10) and (10-11) planes existed in the SLM samples. The EBM sample shows high intensities at (10-10) plane, indicating the presence of texture of (10-10) plane. While the forging sample shows a relatively random distribution of planes.
The manuscript was revised accordingly.
3. What do "Black lines" at 118 lines mean ? You should need to explain further.
Response: The black lines was explained further in the revised manuscript.
4. What does the under line of "cyclic load" at 155 lines mean ?
Response: the words "under cyclic load" were corrected to “during tribological testing with a load of 50 N” in the revised manuscript.
5. I can not find Fig.7 (f, i) at 156 lines.
Response: it is Fig. 7 (a-c). In the revised manuscript, we changed the words “which result in the dimples observed in Fig. 7 (f,i)” to “which result in ploughing grooves observed in Fig. 6 (d, g)”.
6. What is "tribo-layer" at 159 lines ? You should need to explain further.
Response: to explain further, we performed XRD test for the samples after tribological testing. Fig. 2b shows the XRD analysis of the tribo-layer. It shows that the diffraction peaks are more corresponding to hexagonal close-packed (hcp) α phase rather than martensite α′ phase, indicating that martensite phase undergoes transformation to α phase during the test. Meanwhile, it reveals that the texture of (10-10) and (10-11) in the SLM sample and texture of (10-10) plane in the EBM sample disappeared or significantly weakened after testing, while the peak of (0002) plane for all the samples becomes much stronger showing that the texture of (0002) plane occurred during tribological testing. In addition, all the peaks become wider indicating added residual strain in these samples.
7. I can not find Fig.8c at 165 lines.
Response: It is Fig. 8b, which was corrected in the revised manuscript.
8. What does the arrow showed in Fig.8a mean ?
Response: The arrow was deleted in the revised manuscript.
9. Fig.9 does not explain any wear mechanisms. For example, you should explain the effects of the carbides originated from the steel ring and AM process by using Fig.9.
Response: Fig. 9 is trying to show the different crack propagation behavior in the tribo-layer of the samples fabricated by different processes, we use “crack propagation behavior in the tribo-layer” to replace the words of “wear mechanisms” in the revised manuscript. The carbides are not shown in the schematic figure 9 because it is not surely confirmed by the results.
Round 2
Reviewer 1 Report
I accept the article in present form.